# Avidin-biotin complex-based capture coating platform for universal *Influenza virus* immobilization and characterization

**Micaela Trexler**[1,2]*, **Michelle Brusatori**[1,2,3], **Gregory Auner**[1,2,3]

**1** Smart Sensors and Integrated Microsystems, Wayne State University, Detroit, Michigan, United States of America, **2** Department of Biomedical Engineering, Wayne State University College of Engineering, Detroit, Michigan, United States of America, **3** Michael and Marian Illitch Department of Surgery, Wayne State University School of Medicine, Detroit, Michigan, United States of America

* mtrexler@wayne.edu

**Data Availability Statement:** All relevant data are within the manuscript and its Supporting Information files.

**Funding:** M.T., M.B., and G.A.'s work in this study was funded by the Paul U. Strauss endowed chair

## Abstract

Influenza virus mutates quickly and unpredictably creating emerging pathogenic strains that are difficult to detect, diagnose, and characterize. Conventional tools to study and characterize virus, such as next generation sequencing, genome amplification (RT-PCR), and serological antibody testing, are not adequately suited to rapidly mutating pathogens like Influenza virus where the success of infection heavily depends on the phenotypic expression of surface glycoproteins. Bridging the gap between genome and pathogenic expression remains a challenge. Using sialic acid as a universal Influenza virus binding receptor, a novel virus avidin-biotin complex-based capture coating was developed and characterized that may be used to create future diagnostic and interrogation platforms for viable whole Influenza virus. First, fluorescent FITC probe studies were used to optimize coating component concentrations. Then atomic force microscopy (AFM) was used to profile the surface characteristics of the novel capture coating, acquire topographical imaging of Influenza particles immobilized by the coating, and calculate the capture efficiency of the coating (over 90%) for all four representative human Influenza virus strains tested.

## Introduction

Influenza virus mutates quickly and unpredictably creating emerging pathogenic strains that are difficult to detect, diagnosis, and characterize. Each year there are millions of flu cases and tens of thousands of deaths in the United States alone [1]. The wide variety of circulating strains of Influenza virus at any given time contributes to the difficulty of selecting candidate strains for yearly seasonal flu vaccine development as well as making the study of Influenza virus as a species challenging. Strains may suddenly emerge through antigenic shift (reassortment) and become pandemic, such as the 1918 H1N1 "Spanish flu", 1957 H2N2 "Asian flu", or 2009 H1N1 "swine flu" [2]. These pandemic strains often remain circulating in the population for decades, undergoing gradual antigenic drift to cause milder, yearly epidemics until they completely disappear from circulation.

position and the Smart Sensors and Integrated Microsystems Program (http://www.ssim.wayne.edu/index.asp) at the Wayne State University School of Medicine (https://www.med.wayne.edu/). The funders had no role in study design, data collection and analysis, decision to publish, or preparation of the manuscript.

**Competing interests:** The authors have declared that no competing interests exist.

Influenza virus relies on two glycoproteins on its enveloped surface to successfully bind to and later release from epithelial cells of the host respiratory tract—hemagglutinin (HA) and neuraminidase (NA). Influenza HA's are known to multivalently bind to sialic acid (SA) residues of the host cell's glycocalyx during the first step of the viral infection cycle [3–5]. This SA binding triggers viral entry through endocytosis and mediates subsequent endosomal membrane fusion resulting in the release of viral ribonucleoprotein into the host cell [6]. One of the major differences among HA subtypes of Influenza virus is the preferential binding to a specific conformation of the sialic acid glycosidic linkage to glycolipids and glycoproteins of the host cell glycocalyx. For instance, HA subtypes of human Influenza A virus preferentially bind to sialic acid with an α2,6 glycosidic linkage which are abundant on the human pulmonary epithelium [7]. Whereas HA subtypes of avian Influenza A strains prefer an α2,3 glycosidic linkage. This preferential receptor binding is likely one of the major barriers preventing the spread of highly pathogenic avian influenza A virus in humans [8]. Conversely, Influenza B is an exclusively human circulating class of Influenza virus that has a much lower mutation rate in the HA encoded region of its genetic material. The Influenza B virus HA contains a conserved, narrower sialic acid binding site that preferentially binds to the α2,6 glycosidic conformation by discriminating against the avian α2,3 glycosidic linkage [6].

However, there is evidence that the sialic acid residue preference of HA is not absolute. Species crossover has been proven to begin before changes in the HA binding domain occurs, such as during the 1997 Hong Kong bird flu outbreak caused by an avian H5N1 influenza A virus [9]. All eight gene segments of the virus were of avian origin and maintained preference for binding the avian α2,3 glycosidic linkage, yet were able to cause significant pathogenicity in the human population. Better tools are needed to elucidate the mechanism of infection for these crossover events. Conventional tools to study and characterize virus, such as next generation sequencing, genome amplification (RT-PCR), and serological antibody testing, are not adequately suited to rapidly mutating pathogens like Influenza virus where the success of infection heavily depends on the phenotypic expression of surface glycoproteins. Bridging the gap between genome and pathogenic expression remains a challenge. Mutations in PCR primer or probe binding regions significantly impact diagnostic sensitivity and often result in false-negative results [10]. And producing and harvesting strain specific antibodies is a labor and time intensive process plagued by low sensitivities [11, 12]. However, these techniques remain the gold standard for Influenza virus diagnosis, providing guidance for current epidemiological tracking and vaccine strain selection. Recent incremental progress on more complex techniques, such as digital electrochemical enzyme-linked immunoassay (ELISA), for Influenza virus which has a low detection limit [13, 14]. In addition, glycan microarrays are employed to determine binding specificities on a diverse set on glycan configurations [15, 16], but the synthesis and/or purification of well characterized oligosaccharides may be time consuming and prohibitively complex for many diagnostic device labs.

There remains a great need for a paradigm shift in approach to influenza virus diagnostic and characterization techniques. The future of single-molecule biosensors depends on high spatial resolution and immobilization of viable virus to achieve enhanced molecular information, dynamic interactions, and detection sensitivity [17]. The development of open platform technologies is sorely needed to allow for the rapid testing of a variety of unconventional tools, from novel antigenic testing to spectroscopy techniques. Some groups have demonstrated the merit of such techniques by immobilizing isolated Influenza virus sourced antigens to study binding affinity and dynamic binding of Influenza HA and NA to sialic acid [18, 19]. Others have used an impedimetric-based detector to differentiate between species of Influenza A virus [20, 21]. Yet, to the best of our knowledge, universal Influenza binding for glycoprotein characterization studies on viable, whole virus has not been realized without the use of serotype

specific antibodies or species-specific galactose residues. These studies would require the capture and immobilization of all Influenza virus present, agnostic to the specific strain or serotype. This has motivated the development of the novel capture coating in this study, allowing for universal Influenza virus immobilization.

In addition, the evolution of the capture coating design was particularly motivated by the advancement of vibrational spectroscopy techniques, such as infrared spectroscopy, Raman spectroscopy, surface enhanced Raman spectroscopy, and tip enhanced Raman spectroscopy. These techniques are capable of detecting molecular-level phenotypic changes such as those that occur during a viral envelope protein mutation. For example, Sun et al. used surface enhanced Raman spectroscopy (SERS) based immunosensing techniques to detect clinically isolated Influenza A down to 10 pfu/mL [22]. For the cleanest spectra, consideration must be given to the potential background signal created by the immobilization technique used. This motivated the pursuit of an avidin-biotin complex-based technique that can be layered onto optical spectroscopy compatible substrates, such as glass and sapphire, for greater flexibility in data acquisition techniques. Furthermore, application directly to glass and similar substrates would facilitate a variety of microfluidic integrations that befit from virus immobilization, including interrogation for virus characterization and diagnosis or high throughput drug screening.

Towards this end, this study demonstrates the successful development of a novel universal Influenza virus capture coating that harnesses sialic acid binding to capture viable whole Influenza virus. A strong base layer of avidin biotin complex (ABC) binding combined with a bio-mimicking pegylated sialic acid tether is utilized to capture and immobilize the virus relative to the substrate in an aqueous environment without compromising the structural or functional integrity of the virus itself. In this study we confirm virus capture using fluorescent probes and atomic force microscopy (AFM).

## Materials and methods

### Reagents and biologics

Biotinylated bovine serum albumin (bBSA), avidin, blocker BSA, and serotype specific anti-Influenza virus FITC conjugated antibody probes were sourced from Thermo Fisher Scientific (PA1-73044, PA1-73047, PA1-73036). The custom assay probes are biotinylated polyethylene glycol (MW 2000) conjugated with sialic acid (bPEG$_{2k}$SA), produced by Nanocs, Inc. The non-binding assay probe control is biotinylated polyethylene glycol (MW 2000) conjugated with thiol (bPEG$_{2k}$SH), also produced by Nanocs, Inc. Virus strains were acquired from ATCC (IAV H1N1 A/Virginia/ATCC1/2009, IAV H3N2 A/Victoria/3/75, B Yamagata B/Wisconsin/1/2010, and B Victoria B/Florida/78/2015), rehydrated and diluted in PBS 1x and stored at -80˚C with glycerol until thawed for experimental use.

### Avidin-biotin sialic acid capture coating fabrication

1000μg/mL bBSA in PBS was adsorbed onto hydrophobic substrate (black-walled non-treated polystyrene microplate or sapphire slide) via incubation at 37˚C for 2 hours. Substrate was then rinsed with PBS and incubated with 40mg/mL blocker BSA in PBS for 1 hour at 37˚C. Substrate was again rinsed and incubated with 100 μg/mL avidin in PBS for 1 hour at 4˚C followed by another rinse and incubation with 10 μM bPEG$_{2k}$SA (or 10 μM bPEG$_{2k}$SH) in PBS for 1 hour at 4˚C. Substrate underwent a final rinse with PBS, sealed, and stored at 4˚C for up to four weeks without noticeable capture efficiency loss.

## Virus preparation

On testing days, virus stock was removed from -80°C storage and thawed in a 37°C water bath. All stocks consisted of PBS-diluted active virus originating from either pooled allantoic fluid or MDCK propagation supernatant sourced from the supplier, ATCC. Prepared substrates were removed from 4°C storage at this time and brought to room temperature. Once thawed, virus stocks were further diluted in PBS as required to achieve final concentration and incubated on capture coated substrate at 37°C for 1 hour followed by a PBS wash.

## Capture coating optimization

Concentrations of bBSA and avidin were systematically varied from 0 μg/mL to 1000μg/mL and bPEG$_{2k}$SA varied from 0 μM to 100 μM to optimize capture coating efficiency to the final concentrations described under Avidin-Biotin Sialic Acid Capture Coating Fabrication (**Table 1** and **S1 Table**). Serotype specific (H1N1, H3N2, B) anti-Influenza virus FITC conjugated antibody probes were used to detect Influenza virus immobilized on the capture coating. Fluorescent images were captured for each well of a black walled microplate during exposure with an X-CITE 120 fluorescent illuminator fitted with a 480 nm excitation filter with a focal point power of 8.1 mW. Emission was imaged with a SPOT Insight camera through a 40x Nikon Plan Fluor objective and SPOTAdvanced software set to a 519 nm monochrome colorizing palette. Relative fluorescent unit measurements were made using ImageJ opensource software on the captured images. Statistical p values shown in **Table 1** and **S1 Fig** were calculated using a Welch's t-test with a threshold of α = 0.05 on the log$_{10}$ transformation of the fluorescence data. Fluorophore-conjugated antibodies have been determined to have intensity measurements that follow a lognormal distribution [23]. The geometric mean was then calculated by taking the antilog of the mean log$_{10}$ transformed data and reported with the geometric standard deviation in **Table 1** and **S1 Fig**.

## Atomic force microscopy

As an additional confirmation of capture coating efficacy, Atomic Force Microscopy (AFM) was used to image Influenza virus immobilized on the capture coating. A double side polished 460μm thick c-plane sapphire wafer (MSE Supplies) was cut into approximately 3/4" square pieces and layered with capture coating according to Avidin-Biotin Sialic Acid Capture Coating Fabrication protocol and incubated with Influenza virus according to Virus Preparation. Following incubation, slides were gently washed with sterile DI water and allowed to dry under the hood. Sapphire slides with immobilized virus were then placed on the stage of a Park Systems XE series AFM and imaged in soft non-contact mode using an approximately

**Table 1. Fluorescence readings from anti-influenza FITC probes reveal the optimal bPEG$_{2k}$SA concentration to be 10 μM.**

| bPEG$_{2k}$SA | Influenza A H1N1 (A/Virginia/ATCC1/2009) | Influenza A H3N2 (A/Victoria/3/75) | Influenza B Yamagata (B/Wisconsin/1/2010) |
|---|---|---|---|
| 100uM | 241.2 **x/÷ 6.6** | 13222.3 **x/÷ 1.2**$^{*}$ | 638.9 **x/÷ 12.2** |
| **10uM** | **15948.3 x/÷ 1.1**$^{*}$ | **16321.4 x/÷ 1.1**$^{*}$ | **11679.1 x/÷ 1.6**$^{*}$ |
| 1uM | 1282.0 **x/÷ 2.6** | -925.8 **x/÷ 2.3** | -54.0 **x/÷ 1.1** |
| 0uM | 0.0 **x/÷ 2.5** | 0.0 **x/÷ 1.6** | 0.0 **x/÷ 1.4** |

All measurements taken on a base of 1000 μg/mL bBSA and 100 μg/mL avidin. Fluorescence values were obtained by subtracting residual fluorescence values from the 0 μM bPEG$_{2k}$SA control and colormetrically scaled. All cells were incubated with either $10^4$ CEID$_{50}$/mL (IAV H3N2 and IBV Yamagata) or $10^4$ PFU/mL (IAV H1N1). n = 3 for each cell with geometric standard deviation shown.

$^{*}$p < 0.05 compared to respective 0 μM bPEG$_{2k}$SA control.

30nm diameter cantilever probe (Applied NanoStructures, Inc.) with Park Systems XE software. AFM images were processed, and influenza virus particles counted and characterized using Gwyddion open source software for SPM data analysis. Each AFM image was leveled using Gwyddion mean plane subtraction, scars corrected, and background subtracted such that the minimum value was set at zero microns. Using the mark grains feature, a threshold was set to 100 nm height and a 70 nm equivalent radius ($r_{eq}$) filter applied to account for the 30nm radius AFM probe tip causing broadening edge artefacts of the typically 40–100 nm radius Influenza virus particles. Particles were counted by their $r_{eq}$ properties. Those with a $r_{eq}$ larger than 150 nm were double counted as a clustered pair of viruses. An example of this process is depicted in S3 Fig. In addition to the $r_{eq}$, height above capture coating was recorded for each virus.

## Results and discussion

### Capture coating development and optimization

The virus capture coating of the immobilization platform (Fig 1) is comprised of $bPEG_{2k}SA$ linked to bBSA with avidin. Concentration combinations of the three coating components (bBSA, avidin, and $bPEG_{2k}SA$) were investigated to find the optimal concentration based on viral binding performance. This was accomplished by first determining effective combinations of bBSA and avidin. Twenty-five concentration combinations of immobilized bBSA and FITC-conjugated avidin were evaluated by fluorescence imaging. Black-walled non-treated polystyrene microplates were used to prevent fluorescence emission from leaking into adjacent wells during imaging. Fluorescence readings for the FITC-biotin probe provided many promising combinations, as shown by high levels of fluorescence (green) in the upper left triangle of the test matrix, S1 Table. Higher fluorescence readings indicate a higher quantity of potential binding sites for the next layer of $bPEG_{2k}SA$ tether. This agrees with other similar ABC based protocols in literature commonly using avidin to biotin ratios ranging from 1:1 to 1:10 [24–26]. Avidin-biotin binding was chosen as the base of our capture coating because of its many

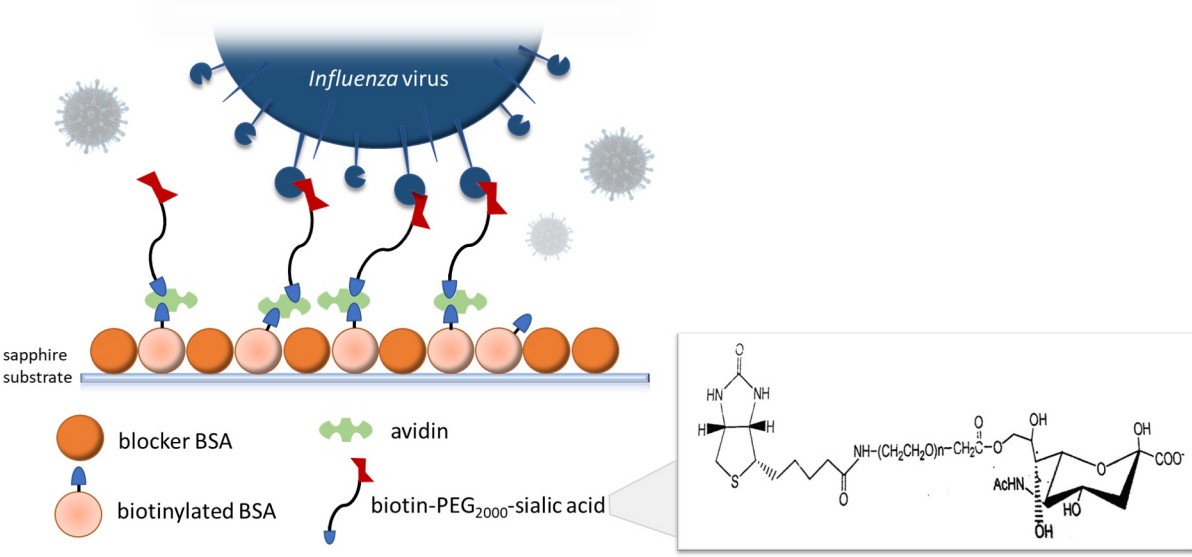

**Fig 1. Capture coating schematic depicting the layers of the custom avidin-biotin complex adsorbed to optically polished, c-cut sapphire slide windows.** The hemagglutinin (HA) glycoproteins on the envelope of the Influenza virus bind sialic acid of the functionalized biotin-PEG linker (structural formula shown, insert, was provided by the supplier, Nanocs, Inc.).

advantageous properties. It is one of the strongest known non-covalent bonds between a protein and ligand with a high degree of affinity ($K_D \approx 10^{-15}$ M) and specificity [27]. Avidin and biotin are widely available reagents with an interaction that is stable over a wide range of temperatures and pH, providing a robust base that would allow for a variety of downstream testing on captured virus.

Next, substrates were prepared with candidate bBSA/avidin combinations and incubated with different concentrations of $bPEG_{2k}SA$. The binding performance of each coating was assessed for three influenza virus strains (A H1N1, A H3N2, and B Yamagata) using serotype specific FITC conjugated anti-influenza probes. The FITC conjugated anti-influenza probes were raised against similar strains as each of the three used in this work (anti-H1N1, anti-H3N2, and anti-B Yamagata). **Table 1** shows the relative fluorescence intensity values for different concentrations of $bPEG_{2k}SA$ with $10^4$ $CEID_{50}$/mL (IAV H3N2 and IBV Yamagata) or $10^4$ PFU/mL (IAV H1N1) of virus. These experimental strains were chosen as representatives of major Influenza serotypes currently endemic in the human population–IAV H1N1, IAV H3N2, and IBV Yamagata, respectively. Results indicate that each of the experimental strains of Influenza virus were best immobilized using 10 μM of $bPEG_{2k}SA$ receptor with 1000 μg/mL bBSA and 100 μg/mL avidin, as indicated by having significantly higher fluorescent reading compared to the 0 μM $bPEG_{2k}SA$ controls (IAV H1N1 p = 0.041, IAV H3N2 p = 0.032, IBV Yamagata p < 0.001). In addition, IAV H3N2 showed significant capture via fluorescence detection at 100 μM $bPEG_{2k}SA$ (p = 0.031), however this was not the case for the other two strains at the same receptor concentration. The high standard deviation of these fluorescence tests, especially at the higher 100 μM $bPEG_{2k}SA$ concentration, may be due to steric hindrance effects caused by a high concentration of receptors. Steric hinderance may be why 10 μM of $bPEG_{2k}SA$ receptor significantly outperformed a higher concentration of 100 μM $bPEG_{2k}SA$ receptor. Other less successful avidin/bBSA combinations with $bPEG_{2k}SA$ are not shown.

Successful immobilization of the representative strains demonstrates proof of concept for this novel capture coating technique to be used universally with Influenza viruses type A and B. Furthermore, detection of Influenza in the concentration demonstrated, $10^4$ PFU/mL or $CEID_{50}$/mL depending on strain used, points toward potential clinical use of this novel immobilization platform. The limit of detection for point of care Influenza virus detection tests has been found to range from 5.4 to 8.9 log copies/mL and 4.8 to 7.3 log copies/mL for Influenza H3N2v and H7N9 viruses strains, respectively [28]. These findings track well with $10^5$ to $10^7$ RNA copies per mL of mean viral load found in human nasopharyngeal isolates [29]. Using an established conversion rate of 20–60 viral genome copies needed per PFU [30], this amounts to a typical clinical range of $10^4$ to $10^6$ PFU/mL in a clinical isolate. A control experiment using an alternative thiol-functionalized $bPEG_{2k}SH$ tether was used to ensure that the Influenza virions were binding to the sialic acid tether and not simply being caught in the tendril-like structures of the capture coating. Results, **S1 Fig**, show consistently lower fluorescence values in this non-binding control coating for all Influenza strains tested compared to the $bPEG_{2k}SA$ tether. Using FITC probes as a quantitative analog to virus concentration was not possible because the different serotype specific anti-Influenza FITC probes used may vary in binding affinity and fluorescence levels across strains.

To our knowledge, this is the first known successful use of the $bPEG_{2k}SA$ molecule in an ABC-based capture coating designed specifically for universal immobilizing infectious virus. Past studies have often used fetuin and/or mucin as a viral receptor, due to an abundance of endogenous sialic acid end chains, in order to study Influenza virus binding characteristics [31–33]. In these cases, the endogenous receptors do not permanently immobilize the virus due to the release mechanism of the vial NA protein. While useful to study binding kinetics, this release mechanism would hinder interrogation by any technique that requires a highly

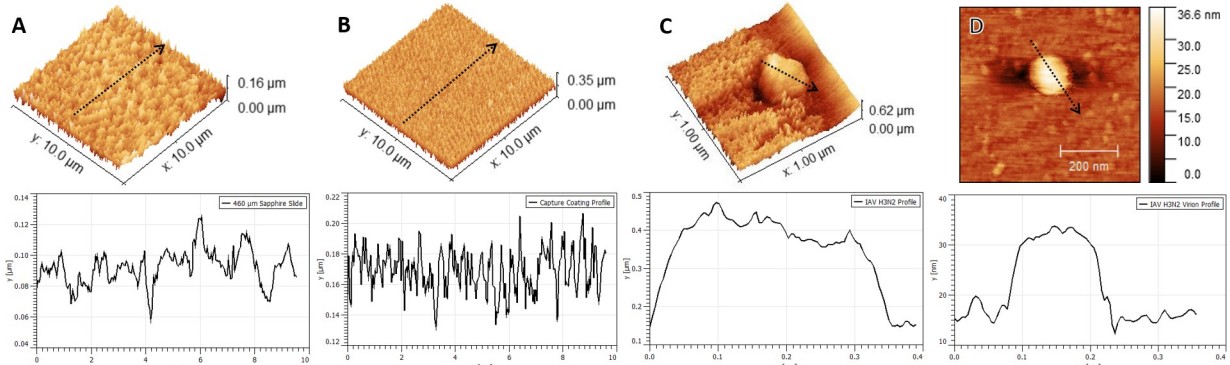

**Fig 2. 3D AFM topography images with corresponding 1D profiles through dashed lines as shown.** A) 460 μm sapphire c-cut slide. B) Capture coating on sapphire slide. C) Influenza A H3N2 virus cluster immobilized in capture coating on sapphire slide. The tendrils of the capture coating bPEG$_{2k}$SA receptor tethers are clearly seen. D) A 2D example image of one of many single virions (this one Influenza A H3N2) immobilized in capture coating. 2DFFT analysis for all four AFM scans shown in **S2 Fig**.

spatially resolved location of the virus, such as super resolution microscopies and spectroscopies. This effect may notably be incompatible in a high laminar flow environment such as a microfluidic. Specifically, in the case of molecular spectroscopies such as Raman spectroscopy and infrared spectroscopy, molecules like fetuin and mucin add an uncontrolled layer of variable molecular makeup that is much more difficult to deconvolve from spectra than well-regulated proteins and molecules like biotin, avidin, and PEG.

A related pegylated sialic acid was recently used to study dynamic binding of HA and NA to sialic acid, including a demonstration of the irreversibility of HA binding in the absence of NA cleaving activity [18], also previously described by Guo et al. [34]. Similarly, our biotinylated polyethylene glycol (MW 2000) conjugated with sialic acid (bPEG$_{2k}$SA) for virus capture was configured without the addition of a galactose-sialic acid linkage to prevent NA cleaving the virus from the pegylated sialic acid tether. This cleaving allows for the severance of progeny virus from the infected host cell *in vivo* [35]. However, it has been shown that this action causes unstable binding of virus *in vitro* as HA mediated binding competes with NA mediated cleaving [36]. Future iterations of this platform may explore incorporating a galactose-sialic acid functionalized end chain with either an α2,6 or α2,3 glycosidic linkage. This may allow for studying the HA mediated binding and NA mediated cleaving of Influenza strains originating from different host species. However, for the scope of this research, only human strains preferential to α2,6 were used, and the glycosidic linkage was not included in the bPEG$_{2k}$SA probe to prevent NA mediated cleaving.

## AFM visualization and analysis

The developed capture coating was characterized using Atomic Force Microscopy (AFM) imaging and topography analysis. While AFM is not practical in a clinical diagnostic setting, it provides valuable validation of Influenza virus capture and allows for characterization of the capture coating itself which lead to the calculation of the coating's *capture efficiency*. **Fig 2** displays topographical images and profile analysis results comparing images of the sapphire substrate (**Fig 2A**), capture coating on sapphire substrate without virus (**Fig 2B**), a cluster of Influenza A H3N2 virions immobilized by capture coating on sapphire substrate (**Fig 2C**), and a single Influenza A H3N2 virion immobilized by capture coating on sapphire substrate (**Fig 2D**). The measured height of the individual virion, 36 nm, and diameter, 160 nm, agree well with the known size of Influenza virus given the relatively large diameter (~30 nm) of the

AFM probe tip causing a broadening edge artefact in the diameter measurement [37–40]. Throughout these studies multiple 50 x 50 μm, 10 x 10 μm, and 1 x 1 μm AFM images were taken. Using Gwyddion software, virus height and radius were taken for each virus captured. These distributions are plotted in **S4 Fig**.

Further, roughness parameters were extracted from these profiles using Gwyddion software and are summarized in **Table 2**. Differences in sapphire substrate and capture coating profiles support successful deposition of the capture coating onto the substrate. The average roughness (Ra) increased by approximately 24 nm which agrees well with the estimated 20–30 nm thickness of the bBSA-avidin-bPEG$_{2k}$SA coating combination. The regularly spaced and spiked profile of the capture coating also indicates successful adsorption of the bBSA and subsequent orientation of the avidin and bPEG$_{2k}$SA binding to create tendril-like protrusions from the substrate surface. These protrusions are better visualized surrounding the AFM image of a cluster of influenza virions (**Fig 2C**) which support the capture coating design and execution first depicted in **Fig 1**. The apparent embedding of the virus into the capture coating is likely due to the brief drying out of the slides before AFM interrogation, causing the PEG tethers to crumple under the weight of the virion. It should be noted that features in the AFM topography images and subsequent roughness parameters may be smaller than they appear due to the relatively large diameter (~30 nm) of the AFM probe tip causing broadening edge artefacts [37].

## Capture efficiency

The capture coating of the virus immobilization platform utilized a unique bPEG$_{2k}$SA linker that provides a biomimicking sialic acid receptor for Influenza HA to bind to. Capture efficiency was calculated for each of the four virus strains. When the total number of virus particles applied to the substrate is less than that which can fully occupy the substrate surface, capture efficiency (C$_{Eff}$) is defined as the ratio of virus particles captured over virus particles applied, **Eq 1**.

$$C_{Eff} = \frac{P_{bound}}{P_{total}} = \frac{S * C}{n * I} \tag{1}$$

Where the number of virus particles bound to the substrate, P$_{bound}$ = S*C with S being the substrate size factor (total substrate area/size of the AFM scan area) and C the virus particle count from AFM image. To determine the virus particle count from an AFM image, a 10 x 10 μm$^2$ AFM scan was analyzed for each of the four Influenza virus strains. Virus particles present were counted using Gwyddion's mark grain feature with minimum thresholds set at 120 nm height and 70 nm equivalent radius to account for the 30nm radius AFM probe tip causing broadening edge artefacts. A demonstration of particle counting via AFM image with Gwyddion mark grain feature is provided as **S3 Fig**. For each Influenza strain, n = 3 AFM images were analyzed for particle count and the mean taken. The total number of virus

**Table 2. Parameters characterizing surface features and roughness of samples from Fig 2.**

| Characterizing Surface Features and Roughness | 460 μm Sapphire Slide | Capture Coating | IAV H3N2 Cluster (ATCC VR-822) | IAV H3N2 Virion (ATCC VR-822) |
|---|---|---|---|---|
| Roughness average (Ra) [nm] | 7.7 | 32.0 | 29.9 | 1.5 |
| Root mean square roughness (Rq) [nm] | 9.2 | 38.1 | 37.1 | 2.0 |
| Maximum roughness height (Rt) [nm] | 43.4 | 191.1 | 186.3 | 9.5 |
| Avg maximum height (Rtm) [nm] | 35.8 | 154.6 | 105.0 | 6.4 |
| Mean spacing of peaks (Sm) [nm] | 588.3 | 333.3 | 49.2 | 45.1 |

**Table 3. Parameters and calculated values of capture efficiency ($C_{Eff}$) for each Influenza strain.**

| Strain | Stock concentration | Volume (mL) | Slide size ($mm^2$) | Mean Particle Count (n = 3) | CTPR [41, 42] (count to PFU ratio) | PtC (PFU to $CEID_{50}$ conversion ratio) | Estimated Capture Efficiency ($C_{Eff}$) |
|---|---|---|---|---|---|---|---|
| IAV H3N2 (A/Victoria/3/75) | $1.6 \times 10^5$ $CEID_{50}$/mL | 0.500 | 1140 | 2.0 ± 1 | 200 | 1.429 | 0.997 (99.7%) |
| IAV H1N1 (A/Virginia/ATCC1/2009) | $2.9 \times 10^4$ PFU/mL | 0.300 | 100 | 5.0 ± 1 | 591 | N/A | 0.972 (97.2%) |
| IBV Yamagata (B/Wisconsin/1/2010) | $2.8 \times 10^5$ $CEID_{50}$/mL | 0.500 | 1140 | 16.0 ± 2 | 1000 | 1.429 | 0.912 (91.2%) |
| IBV Victoria (B/Florida/78/2015) | $7.8 \times 10^5$ $CEID_{50}$/mL | 0.500 | 1140 | 11.0 ± 4 | 250 | 1.429 | 0.900 (90.0%) |

particles applied to the substrate $P_{total} = n*I$, where n is the number density of virus particles in the stock solution and I is the volume of sample applied to the substrate. The number density is calculated from the infectious virus concentration of the virus stock [V] multiplied by a conversion factor. For virus concentration in PFU/ml, n = [V]*CTPR, where CTPR is the virus particle "count to PFU ratio" found in previous studies [41, 42]. When the virus concentration is $CEID_{50}$/ml, n = [V]*PtC*CTPR, where PtC is the PFU to $CEID_{50}$ ratio conversion factor.

Results and inputs parameters used to calculate the *capture coating efficiency* are detailed in **Table 3**. For this work, it is estimated that the total number of virus particles incubated for 1 hour on a functionalized slide ($P_{total} = 2.29 \times 10^7$, $5.14 \times 10^6$, $2.00 \times 10^8$, $1.39 \times 10^8$) was less than the total number of available binding sites ($I_{total} = 1.03 \times 10^{10}$, $9.02 \times 10^8$, $1.03 \times 10^{10}$, $1.03 \times 10^{10}$) for IAV H3N2, IAV H1N1, IBV Yamagata, and IBV Victoria, respectively. Capture efficiency ($C_{Eff}$) ranged from 90.0% to 99.7%. These promising results are likely due to the irreversibility of HA binding to sialic acid in absence of active NA cleaving [18] or due to strong avidity from bivalent [43] or multivalent binding of sialic acid by individual Influenza virions [36, 44–46]. Multivalent binding of Influenza virus to our capture coating may be caused by multiple $bPEG_{2k}SA$ binding to a single avidin molecule in the ABC complex. Alternatively, the 333 nm peak spacing found via the AFM topography data had a standard deviation of 90 nm. The influenza virus may also be showing preference to binding locations where multiple $bPEG_{2k}SA$ binding locations are within reach, providing access for multivalent binding. These results demonstrate the strong immobilization capability of this novel $bPEG_{2k}SA$ capture coating. However, this strong binding capability was generated under ideal, static incubation conditions and should be further vetted through flow cell observations beyond the scope of this study.

Sialic acid is a common binding receptor used by a variety of viruses to initiate host cell entry. Since the goal of this research was to create a universal infectious virus Influenza immobilization platform, the sialic acid functionalized bPEG probe was synthesized without galactose and, therefore, without the NA glycosidic linkage cleaving site. While this work was limited to just a single conformation of $bPEG_{2k}SA$, advanced binding studies may be done by altering the glycan chemistry, such as functionalizing the PEG with a α2,6 glycosidic conformation favored by human pathogenic Influenza virus or, alternatively, an avian Influenza favored α2,3 glycosidic linkage. In addition, further studies may open the possibility to use this capture coating with other SA binding virus families such as adenovirus and coronavirus, amongst others [43, 47]. Like Influenza virus ($K_d$ = μM-pM) [48], adenovirus ($K_d$ = 19 μM) [47], and some coronaviruses ($K_d$ = 49.1 μM) [49] *multivalently* bind to SA with $K_d$ in a high affinity range.

Together, these fluorescence and AFM studies show the first successful use of a novel ABC based sialic acid capture coating for universal Influenza virus immobilization and its potential as a diagnostic tool platform. An avidin-biotin complex base with a biotin-PEG-sialic acid

functionalized surface was utilized to create an Influenza virus capture coating capable of immobilizing whole virus while keeping the virion intact. Atomic Force Microscopy studies confirmed and described the profile of the capture coating as well as demonstrated the ability of using the capture coating to help characterize a single Influenza virus by AFM topography. From AFM topography data we were able to determine the capture efficiency of the capture coating to be above 90% of virus particles at a concentration of $10^5$ $CEID_{50}$/mL. We hope this capture coating technique creates an adaptable platform for further characterization of virus and development of novel phenotype diagnostic techniques.

## Supporting information

**S1 Fig. Fluorescent tagging was used to assess the novel $bPEG_{2k}SA$ capture coating against a non-binding control ($bPEG_{2k}$ functionalized with a thiol group, SH).** A) Schematic of experimental set up. Influenza virus were incubated on capture coating or control coating, rinsed, and tagged using FITC conjugated anti-Influenza HA antibodies. B) Fluorescence results in relative fluorescence units (RFU). Numbers reported as geometric mean and geometric standard deviation (GSD) ($^*$ = $p < 0.05$, $^{**}$ = $p < 0.01$, $^{***}$ = $p < 0.001$).
(TIF)

**S2 Fig. Corresponding 2D FFT analysis on AFM images from Fig 2.** A) 460 μm c-cut sapphire slide. Image suggest spatial features of a repeated, relatively smooth texture. B) Capture coating on sapphire slide. Image suggests spatial features of a repeated, dotted, and quite messy texture. C) Clump of Influenza A H3N2 virion immobilized by capture coting on sapphire slide. The image suggests the virion has a gaussian-like shape with a randomly rough texture within a small variation of the larger gaussian surface. Similarly, D) shows a single Influenza A H3N2 virion immobilized by capture coating on sapphire slide. All 2D FFT analysis was conducted using Gwyddion open source SPM analysis software on captured AFM.tiff images.
(TIF)

**S3 Fig. Virus particle counting for $C_{Eff}$ calculations was carried out using the Gwyddion SPM analysis software.** A 10 μm x 10 μm AFM image of immobilized $10^5$ $CEID_{50}$/mL IBV Yamagata on capture coating was imported processed. (A)The image was leveled using Gwyddion mean plane subtraction, scars corrected, and background subtracted such that the minimum value was set at zero microns. (B) Using the mark grains feature, a threshold was set to 100 nm height and a 70 nm equivalent radius ($r_{eq}$) filter applied to account for the 30nm radius AFM probe tip causing broadening edge artefacts of the typically 40–100 nm radius Influenza virus particles. The mask is colored in pink against a grey excluded background. (C) Particles were counted by their $r_{eq}$ properties. Those with a $r_{eq}$ larger than 150 nm were double counted as a clustered pair of viruses. This image resulted in a count of 16 IBV Yamagata particles in the given frame.
(TIF)

**S4 Fig. Height and radius distributions of virus immobilized by capture coating.** The mean radius and height of each strain was not significantly different than each other strain and were as follows (height, radius): IAV H1N1 (34.16 nm, 42.98 nm), IAV H3N2 (57.75 nm, 53.52 nm), IBV Victoria (41.44 nm, 47.78 nm), and IBV Yamagata (37.68 nm, 51.59 nm).
(TIF)

**S1 Table. Fluorescence readings from FITC-biotin probes used to screen for acceptable bBSA and avidin concentration combinations used for further capture coating testing.**

Fluorescence values were obtained by subtracting residual fluorescence values from the 0 μg/mL bBSA and 0 μg/mL avidin control and colormetrically scaled. Fluorescent images were captured for each well of a black walled microplate during exposure with an X-CITE 120 fluorescent illuminator fitted with a 480 nm excitation filter with a focal point power of 8.1 mW. Emission was imaged with a SPOT Insight camera through a 40x Nikon Plan Fluor objective and SPOTAdvanced software set to a 519 nm monochrome colorizing palette. Fluorescent unit measurements were made using ImageJ opensource software on the captured images. (TIF)

## Acknowledgments

The authors would like to acknowledge the contributions of Dr. Gina Shreve and Chris Thrush.

## Author Contributions

**Conceptualization:** Micaela Trexler, Gregory Auner.

**Data curation:** Micaela Trexler.

**Formal analysis:** Micaela Trexler, Michelle Brusatori.

**Funding acquisition:** Gregory Auner.

**Investigation:** Micaela Trexler.

**Methodology:** Micaela Trexler, Gregory Auner.

**Resources:** Gregory Auner.

**Supervision:** Gregory Auner.

**Validation:** Micaela Trexler.

**Visualization:** Micaela Trexler.

**Writing – original draft:** Micaela Trexler.

**Writing – review & editing:** Micaela Trexler, Michelle Brusatori, Gregory Auner.

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
