## [Decision Letter · Decision Letter 0]

24 Dec 2020

PONE-D-20-36336

Avidin-Biotin Complex Based Capture Coating Platform for Universal Influenza Virus Immobilization and Characterization

PLOS ONE

Dear Dr. Trexler,

Thank you for submitting your manuscript to PLOS ONE. After careful consideration, we feel that it has merit but does not fully meet PLOS ONE’s publication criteria as it currently stands. Therefore, we invite you to submit a revised version of the manuscript that addresses the points raised during the review process.

Reviewer one questions the novelty. Please emphasize on the originality. Reviewer 2 is concerned about the AFM images and wonder if the particules are really virus. Please take care to consider this comment carefully.

We look forward to receiving your revised manuscript.

Kind regards,

Etienne Dague, PhD

Academic Editor

PLOS ONE

2. Our staff editors have determined that your manuscript is likely within the scope of our Call for Papers on Influenza. This editorial initiative is headed by PLOS ONE Guest Editors Dr. Meagan Deming and Dr. Deshayne Fell. The Collection encompasses research on influenza prevention on every level, including in vitro, translational, behavioral, and clinical studies; disease and immunity modelling; as well as new approaches to influenza prevention. Additional information can be found on our announcement page: https://collections.plos.org/call-for-papers/influenza/.

Currently, your manuscript is included in the group of papers being considered for this call. Please note that being considered for the Collection does not require additional peer review beyond the journal’s standard process and will not delay the publication of your manuscript if it is accepted by PLOS ONE. We would greatly appreciate your confirmation that you would like your manuscript to be considered for this Collection by indicating this in your next cover letter. If you would prefer to remove your manuscript from collection consideration, please specify this in your cover letter.

3. To comply with PLOS ONE submission guidelines, in your Methods section, please provide additional information regarding your statistical analyses. In addition, please report your p-values to support your claims. For more information on PLOS ONE's expectations for statistical reporting, please see https://journals.plos.org/plosone/s/submission-guidelines.#loc-statistical-reporting.

Reviewers' comments:

Reviewer's Responses to Questions

**Comments to the Author**

1. Is the manuscript technically sound, and do the data support the conclusions?

Reviewer #1: Partly

Reviewer #2: No

2. Has the statistical analysis been performed appropriately and rigorously? 

Reviewer #1: No

Reviewer #2: No

3. Have the authors made all data underlying the findings in their manuscript fully available?

Reviewer #1: Yes

Reviewer #2: Yes

4. Is the manuscript presented in an intelligible fashion and written in standard English?

Reviewer #1: Yes

Reviewer #2: Yes

5. Review Comments to the Author

Reviewer #1: This manuscript reports influenza virus capturing system using avidin-biotin complex and sialic acid (SA). The authors optimized the surface functionalization protocol and suggested the evidences (AFM and fluorescence) of virus capturing. SA is a well-known universal receptor for influenza viruses and avidin-biotin interaction has been widely used in biological experiments. Although the authors claimed that this is the first known successful use of the bPEG2kSA molecule in an ABC-based capture coating designed specifically for immobilizing infectious virus, I think the novelty of this work is insufficient to publish in PLOS one.

1. What is the main purpose of this system? For the diagnosis of influenza viruses, there are several methods. I should be compare the current method with pervious approaches. If the development of surface functionalization method is the key achievement, it will be nice to apply the method to various sensing surfaces.

2. In Table 1, why the fluorescence signals are so irreproducible? Error data is even higher than the average values.

3. For the calculation of capturing efficiency, the related AFM data should be provided.

Reviewer #2: The authors provide an interesting work about capturing flu virus. The topic is appealing an interesting, but I found the AFM part a little weak.

In particular, figure 2c pretends to be an individual virus particle. However, the height provided by the topographical profile reaches almost 500 nm. It is known that influenza virus is a pleomorphic virus whose diameter is around 100 nm (Biophysical Journal (2014) 106(7) 1447–1456), but the AFM topographical height ranges between 60nm and 120 nm. In fact, viruses height on the surface can be more or less collapsed due to the virus-surface interaction (Current Opinion in Virology Volume 18, June 2016, Pages 82-88, Seminars in Cell & Developmental Biology Volume 73, January 2018, Pages 199-208). I wonder if what they show in this figure is a virus. The authors should provide more statistics about the AFM data, such as size and height distributions.

Supplementary figure S3 shows a region with several viruses. How is the statistics of virus heights on the surface? How many individual viruses are like this shown in figure 2?

6. PLOS authors have the option to publish the peer review history of their article (what does this mean?). If published, this will include your full peer review and any attached files.

Reviewer #1: No

Reviewer #2: No

---

## [Author Response · Author response to Decision Letter 0]

20 Jan 2021

The following is copied from the enclosed form "Response to Reviewers". Thank you again for your time, consideration, and insightful comments and recommendations that have led to the improvement of our manuscript. 

Dear Dr. Dague and Reviewers,

Thank you for giving us the opportunity to submit a revised draft of the manuscript “Avidin-Biotin Complex Based Capture Coating Platform for Universal Influenza Virus Immobilization and Characterization” for publication in PLOS ONE. We are grateful of the time and effort that you and the reviewers have dedicated to providing feedback on our manuscript and are appreciative of the insightful comments which have contributed to valuable improvements to our paper. We have incorporated many of the suggestions made by the reviewers and have supplemented our analysis and discussion with emphasis on originality and AFM evidence for the presence of virus. 

Please see below for a point-by-point response to the reviewers’ comments and concerns. All line numbers refer to the revised manuscript with tracked changes. 

Sincerely,

Micaela Trexler, M.Eng 

Doctoral Candidate

Biomedical Engineering

Wayne State University

Detroit, MI 48202

Reviewers’ Comments to the Authors:

Reviewer 1

1. Comment: What is the main purpose of this system? For the diagnosis of influenza viruses, there are several methods. Compare the current method with pervious approaches. 

Author Response: Thank you for pointing out the weakness in our claims on originality and for giving us the opportunity to emphasize the novelty and purpose of this work, especially as it compares to current and previous approaches. We have addressed this concern on multiple fronts.

On the comparison to current diagnostic techniques and how our capture coating may integrate into a novel diagnostic set up, we have added clarifying language and strategic emphasis on the adaptable potential of this work to second half of the Introduction section, lines 70 to 117. These paragraphs detail the pros and cons of existing diagnostic techniques (PCR, ELISA, and glycan microarrays) and the progress, potential, and, up to this point, difficulty of molecular spectroscopy techniques that require universal Influenza virus immobilization. Studies using our novel bPEG2kSA capture coating in conjunction with nano-resolution vibrational spectroscopy to differentiate between Influenza virus strains are ongoing, but we felt are outside the scope of this stand-alone manuscript describing the capture coating itself. 

Regarding the novelty of the capture coating itself, we have addressed this in a new paragraph in the Results and Discussion section, lines 228 to 238. Here we reference previous studies by Lai et. al (2019), Onkhonova et. al (2019), and Sakai et. al (2017) that used fetuin or mucin as a common method of studying virus binding characteristics and why these endogenous receptors would not be ideal for our future diagnostic purposes:

“Past studies have often used fetuin and/or mucin as a viral receptor, due to an abundance of endogenous sialic acid end chains, in order to study Influenza virus binding characteristics[31-33]. In these cases, the endogenous receptors do not permanently immobilize the virus due to the release mechanism of the vial NA protein. While useful to study binding kinetics, this release mechanism would hinder interrogation by any technique that requires a highly spatially resolved location of the virus, such as super resolution microscopies and spectroscopies. This effect may notably be incompatible in a high laminar flow environment such as a microfluidic. Specifically, in the case of molecular spectroscopies such as Raman spectroscopy and infrared spectroscopy, molecules like fetuin and mucin add an uncontrolled layer of variable molecular makeup that is much more difficult to deconvolve from spectra than well-regulated proteins and molecules like biotin, avidin, and PEG.”

In addition, in the following paragraph lines 239 to 243, we reference the only previous study (Guo et. al 2018) we are aware of that uses a pegylated sialic aid conjugate for Influenza virus immobilization. We use this study as support for the receptor binding functionality of our bPEG2kSA end chain, though our study goals are different. 

2. Comment: In Table 1, why is the fluorescence signal error data so high?

Author Response: We agree with your assessment that the standard deviation of our fluorescence experiments summarized in Table 1 is high and have addressed potential reasons with the addition of lines 206 to 209 in the revised manuscript:

“The high standard deviation of these fluorescence tests, especially at the higher 100 µM bPEG2kSA concentration, may be due to steric hindrance effects caused by a high concentration of receptors. Steric hinderance may be why 10 µM of bPEG2kSA receptor significantly outperformed a higher concentration of 100 µM bPEG2kSA receptor.” 

Reflection on this also motivated us to solidify a claim of significance with new statistical analysis (Welch’s t-test, described in Methods section, lines 152 to 156), which required handling the statistic calculations with data from the lognormal distribution typical of fluorescence tagged data. This allowed us to claim the optimized bPEG2kSA concentration of 10 µM had “significantly higher fluorescent reading compared to the 0 µM bPEG2kSA controls (IAV H1N1 p =0.041 , IAV H3N2 p = 0.032, IBV Yamagata p < 0.001).” (revised manuscript lines 203-204) despite the high standard deviation of the data. Values that met the significance criteria are now marked with a “*” in Table 1 and S1 Fig. To reflect these changes, we report the data as geometric mean and standard deviation. 

3. Comment: For the calculation of capturing efficiency, the related AFM data should be provided.

Author Response: Thank you for this observation that made us realize our mistake in erroneously omitting the proper standard deviation data in the “Mean Particle Count” column of Table 3 which is used to calculate the capture efficiency. This data has been added. Further improvements have been made to the AFM section of this manuscript in regard to additional analysis, characterization, and visualization of individual viruses and the capture coating. In the interest of brevity, please refer to our responses to the specific requests of Reviewer 2. 

Reviewer 1, thank you again for your invaluable insight and questions pertaining to the originality and data behind this work. We hope we have satisfactorily addressed your concerns on the novelty of our bPEG2kSA and ABC-based capture coating and readily invite additional feedback as you see fit. 

Reviewer 2

1.Comment: The dimensions of the virus particle in Fig 2c does not match established reference work. 

Authors Response: We agree with your assessment, especially as it pertains the topographical profile of the sample in Fig 2C that we had erroneously claimed to be a single virion. In response to your insight, we thoroughly reviewed our many AFM images taken over the course of our experiments and determined this image to be a cluster of Influenza viruses, not a single virion. We have added Fig 2D to our revised manuscript which provides evidence of a single virion immobilized by the capture coating with a height of 36 nm and a diameter of 160 nm that better matches the reference studies you provided, given the probable collapsed height due to virus-surface interactions and the edge broadening artefacts of our 30 nm AFM probe. These corrections, with suggested references (de Pablo et. al 2017, Li et. al 2014, and Marchetti et. al 2016), have been added to the revised manuscript, both in a revised Fig 2 and in lines 255 to 261: 

“Fig 2 displays topographical images and profile analysis results comparing images of the sapphire substrate (Fig 2A), capture coating on sapphire substrate without virus (Fig 2B), and a cluster of Influenza A H3N2 virions immobilized by capture coating on sapphire substrate (Fig 2C), and a single Influenza A H3N2 virions immobilized by capture coating on sapphire substrate (Fig 2D). The measured height of the individual virion, 36 nm, and diameter, 160 nm, agree well with the known size of Influenza virus given the relatively large diameter (~30 nm) of the AFM probe tip causing a broadening edge artefact in the diameter measurement. [37-40] .”

In addition, we added a new column to Table 2 with the corresponding characterizing surface parameters for the individual virion in Fig 2D. We feel it is important to keep Fig 2C as it provides a wonderful visual of the tendril-like structure of the capture coating. However, now it is properly and clearly described as capturing a cluster of viruses as to not mislead the reader. 

2. Comment: How many individual viruses are like this shown in figure 2?

Author Response: The vast majority of the virions captured by our coating and imaged using AFM closely resembled the newly referenced profiles in conjunction with Fig 2D with heights in the 20 to 60 nm range and radii in the 35 to 70 nm range. We provided evidence for this in response to your next comment below. In all, we were able to evaluate the height and radius dimensions of 218 particles of Influenza A H1N1, 88 of Influenza A H3N2, 93 of Influenza B Victoria lineage, and 42 of Influenza B Yamagata lineage. 

3. Comment: The authors should provide more statistics about the AFM data, such as size and height distributions. Supplementary S3 Fig shows a region with several viruses. How is the statistics of virus heights on the surface? 

Author Response: Prompted by this suggestion, we took advantage of the mark grain feature in the Gwyddion software, as used in the capture coating efficiency related particle counts detailed in S3 Fig, and applied the same applicable thresholds to all AFM images captured over the course of our various experiments. This included all the 10 x 10 µm scans used in the Ceff calculations (including the one shown in S3 Fig), gross 50 x 50 µm scans, and detailed 1 x 1 µm scans. Radius and height for each marked grain (i.e. virus particle) were gathered using Gwyddion’s distributions of various grain characteristics feature and summarized in the histogram plots of the new S4 Fig for each of the four Influenza stains used throughout our experiments. 

Reviewer 2, thank you again for your insight and for challenging us to look closer at our AFM data. We believe these changes have improved our manuscript greatly and hope we have addressed your concerns aptly. We gladly invite additional feedback as needed.

---

## [Decision Letter · Decision Letter 1]

8 Feb 2021

Avidin-biotin complex-based capture coating platform for universal Influenza virus immobilization and characterization

PONE-D-20-36336R1

Dear Dr. Trexler,

We’re pleased to inform you that your manuscript has been judged scientifically suitable for publication and will be formally accepted for publication once it meets all outstanding technical requirements.

Kind regards,

Etienne Dague, PhD

Academic Editor

PLOS ONE

Additional Editor Comments (optional):

Reviewers' comments:

Reviewer's Responses to Questions

**Comments to the Author**

1. If the authors have adequately addressed your comments raised in a previous round of review and you feel that this manuscript is now acceptable for publication, you may indicate that here to bypass the “Comments to the Author” section, enter your conflict of interest statement in the “Confidential to Editor” section, and submit your "Accept" recommendation.

Reviewer #1: All comments have been addressed

Reviewer #2: All comments have been addressed

2. Is the manuscript technically sound, and do the data support the conclusions?

Reviewer #1: Yes

Reviewer #2: Partly

3. Has the statistical analysis been performed appropriately and rigorously? 

Reviewer #1: Yes

Reviewer #2: I Don't Know

4. Have the authors made all data underlying the findings in their manuscript fully available?

Reviewer #1: Yes

Reviewer #2: Yes

5. Is the manuscript presented in an intelligible fashion and written in standard English?

Reviewer #1: Yes

Reviewer #2: Yes

6. Review Comments to the Author

Reviewer #1: The authors's responses solved the concerns of reviewers

I think that the manuscript now suitable for PLOS ONE

Reviewer #2: The authors have replied to my questions.

However, figure 2 is complicated. The font size of the charts is very narrow and difficult to read. I am still missing some statistical AFM data analysis in the main paper.

7. PLOS authors have the option to publish the peer review history of their article (what does this mean?). If published, this will include your full peer review and any attached files.

Reviewer #1: No

Reviewer #2: No

---

## [Editor Report · Acceptance letter]

19 Feb 2021

PONE-D-20-36336R1 

Avidin-biotin complex-based capture coating platform for universal *Influenza virus* immobilization and characterization 

Dear Dr. Trexler:

I'm pleased to inform you that your manuscript has been deemed suitable for publication in PLOS ONE. Congratulations! Your manuscript is now with our production department. 

Kind regards, 

on behalf of

Dr. Etienne Dague 

Academic Editor

PLOS ONE